



# Vertical distribution of black carbon and its mixing state in urban boundary layer in summer

Hang LIU[1], Xiaole PAN[1], Shandong LEI[1,2], Yuting ZHANG[1,2], Aodong Du[1,2], Weijie YAO[1,2], Tao WANG[1],  Jinyuan XIN[2,4,5], Jie LI[1,2], Yele SUN[1,2], Junji CAO[4], Zifa WANG[1,2,3]

[1] State Key Laboratory of Atmospheric Boundary Layer Physics and Atmospheric Chemistry, Institute of Atmospheric Physics, Chinese Academy of Sciences, Beijing, 100029, China

[2] University of Chinese Academy of Sciences, Beijing, 100049, China

Institute of Urban Meteorology, China Meteorological Administration, Beijing, 100089, China

[3] Center for Excellence in Regional Atmospheric Environment, Chinese Academy of Science, Xiamen, 361021, China

[4] Institute of Atmospheric Physics, Chinese Academy of Sciences, Beijing, 100029, China

[5]Collaborative Innovation Center on Forecast and Evaluation of Meteorological Disasters, Nanjing University of Information Science and Technology, Nanjing 210044, China

*Correspondence to*: Xiaole Pan (panxiaole@iap.ac.cn)

**Abstract.** The vertical distribution of black carbon (BC) as well as its mixing state is of great concern due to BC's strong regional climatic and environmental effects. In this study, vertical measurements were conducted through a moveable container based on a meteorology tower in an urban area. A total of 112 vertical profiles (0-240 m), including the concentrations of BC, $O_3$, $NO_x$ and the optical properties of aerosols, were obtained. Based on BC concentration, the vertical profiles could be classified into four categories: uniform, gradual decrease, sharp decrease, and sudden increase. The uniform type indicates

strong vertical mixing with similar pollutant concentrations along the vertical direction. The gradual/sharp decrease types indicate stable vertical conditions with higher pollutant concentrations on the ground and lower concentrations at higher altitudes. Due to the strong radiation in summer, the vertical profiles exhibited a clear diurnal variation in which ~80% of profiles were uniform during the daytime and ~40%-90% of profiles were gradual/sharp decrease types at night. $O_3$ is an exception, and its concentration generally increases with height even under strong vertical mixing conditions. The size

distribution of BC core varied slightly along the vertical direction, and the coating thickness, denoted by the shell/core ratio ($D_p/D_c$), of BC increased with height under stable conditions. Although the coating thickness could increase the absorption ability with an average absorption enhancement of 1.25 at 23:00, the vertical difference of $D_p/D_c$ (2%) was much lower than that of BC concentration (~35%). The vertical variation of absorption ability was mainly caused by the variation of BC concentration. In addition, $O_3$ and $D_p/D_c$ occasionally increased during 6:00-8:00 but remained stable during 8:00-10:00.

Vertical mixing and transportation from upper heights, such as the residual layer, could significantly influence the pollutant



properties on the surface during early mornings. This study exhibits a continuous vertical picture of BC and its mixing state in urban areas, which would be helpful for understanding BC's regional environmental effect.

**Introduction**

Environmental effects associated with black carbon (BC) has attracted much attention (Ramanathan and Carmichael,
2008;Bond et al., 2013;Li et al., 2022a). In addition to being one of the most toxic components of atmospheric pollutants (Dominguez-Rodriguez et al., 2015;Lin et al., 2011;Xue et al., 2021), BC could influence the boundary layer height, which can aggregate air pollution (Ding et al., 2016). Such depressing of the boundary layer is called the "dome effect" in which the BC in the upper boundary layer could heat the area, increasing the stability of the boundary layer. Through unmanned aerial vehicle measurements, Wilcox et al. (2016) found turbulent kinetic energy reduction with increasing BC concentration leading
to a shallower boundary layer. Further studies demonstrated that the dome effect is relevant to the vertical distribution of BC (Wang et al., 2018b). BC could even be attributed to the increase in the boundary layer if the BC concentration in the surface layer was too high. Thus, the precise measurement of BC vertical distribution is essential to evaluate its environmental effect. The present study mainly focused on the vertical distribution of BC mass concentrations (Wang et al., 2018a;Samad et al., 2020;Wu et al., 2021a;Guan et al., 2022). For instance, Lu et al. (2019) found that the vertical profile of BC mass concentration
could be classified into four types, and the variation in the vertical profile was related to the boundary layer variation and local emissions. Based on the measured BC vertical concentration, the precise evaluation of BC's radiative property distribution requires additional BC related information, such as size distribution and mass absorption cross section, which is a property that is commonly assumed in previous studies (Wu et al., 2021a). In fact, the optical properties could be significantly influenced by BC's microphysical properties. For example, the coating material could increase the absorption ability through the lensing
effect. Shiraiwa et al. (2010) found that BC's absorption ability after coating could reach two times greater than BC without coating. In addition, the size distribution and morphology of BC have shown to be important factors related to BC's optical and radiative properties (He et al., 2015;Matsui et al., 2018;Luo et al., 2022;Liu et al., 2020c). Such microphysical properties of BC vary greatly from the reported ground-based measurements (Shiraiwa et al., 2008;Pan et al., 2017;Liu et al., 2020b;Zhao et al., 2022), generally depending on the varied emission sources and complex aging processes. However, vertical
measurements of such microphysical properties are still rare.

The present vertical measurement approaches include tower, balloon, aircraft, and unmanned aerial vehicle (UAV) approaches. UAVs have the advantages of high flexibility, but large precise instruments cannot be loaded by UAVs due to their low carrying capacity. UAV measurements generally focus on mass concentrations by using light instruments or sensors (Liu et al., 2020a;Lu et al., 2020;Pikridas et al., 2019;Kwak et al., 2020). Aircraft is another vertical measurement approach that combines
flexibility and preciseness. Limited vertical measurements of BC and its mixing state have been conducted (Schwarz et al., 2017;Ding et al., 2019). However, the data points in the boundary layer were limited in aircraft measurements due to flight height limitations. In addition, continuous aircraft vertical measurements are difficult to conduct due to cost issues. The balloon

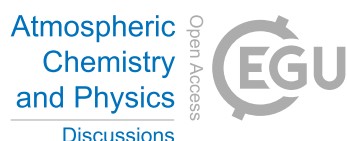

measurements (Liu et al., 2012;Wang et al., 2022) have requirements for the location, which is typically conducted in rural or remote areas. Among the measurement methods, tower measurements are the most suitable way to study the vertical

distribution of BC microphysical characteristics as well as its relationship with the boundary layer in urban areas considering the cost, accuracy, and observation continuity.

BC emissions are substantial in urban areas (Li et al., 2017), exerting a significant impact on the urban environment and climate. The microphysical properties of BC may be highly variable under the influence of fresh emissions, aging and regional transportation. Due to the advantages of high data continuity, low cost and high observation frequency, a tower-based

observation campaign was conducted in Beijing during summer. The vertical distributions of BC concentration, size distribution and mixing state were measured. In addition, aerosol absorption, extinction, and pollutant gases ($CO$, $NO_2$ and $O_3$) were measured simultaneously. Such continuous vertical measurements of BC microphysical characteristics in the urban boundary layer are rare. This research provides insight into the vertical distribution of BC and its properties in the boundary layer and is helpful to reduce the uncertainties in BC environmental and climatic evaluations.

**2. Method**

**2.1 Measurements**

The observation was conducted on the tower campus of the Institute of Atmospheric Physics, Chinese Academy of Sciences (longitude: 116.37°E; latitude: 39.97°N) from June 17 to July 16, 2022. The site is located in the urban area in Beijing and is approximately 50 m (380 m) away from the closet road (highway). More detailed descriptions of the observation site can be

found in the previous literature (Sun et al., 2016;Pan et al., 2017;Xie et al., 2019). All instruments were placed in a moveable container, and an air conditioner and a UPS were used to supply stable temperature and power for the instruments inside. The container is able to move along the cable supporting the meteorology tower. During a normal run of a vertical observation, the container started at the ground level and rose at a near constant speed (8 m/s) to the 240 m height of the tower, stayed at 240 m for 45 minutes and descended to the ground at a speed of 8 m/s. The routine vertical observation was conducted 4 times a

day (7:00, 11:00, 18:00 and 22:00), representing the morning, noon, afternoon, and night conditions of the atmosphere. The vertical observation was cancelled during the times with heavy rain or high wind, and the moveable container stayed at the ground when the vertical observation was not conducted. During the whole campaign, 112 vertical profiles were obtained, and 105 vertical profiles were analysed through data quality control.

Pollutant gases ($NO_2$, $O_3$, $CO$), BC concentration/mixing state and aerosol optical properties (light extinction and absorption)

were measured through the instruments inside the container. In addition to the container, meteorological parameters, including temperature (T), relative humidity (RH), wind speed (WS) and wind direction (WD), at 15 different heights were obtained through the instruments equipped on the tower. All instrument models and manufacturers are listed in Table S2.





## 2.2 Data analysis

### 2.2.1 Coating thickness and size distribution of BC

The BC microphysical properties were obtained through a single particle soot photometer (SP2, Droplet Measurement Technology, Inc., Boulder, CO, USA). SP2 obtained the mass equivalent diameter ($D_c$) of the BC core in every single BC-containing particle, and the optical diameter ($D_p$) of BC-containing particles was derived through the Leading edge only (LEO) fitting method (Gao et al., 2007) and a look-up table (table of precalculated scattering cross section of BC-containing particles with variable $D_c$ and $D_p$ through Mie scattering theory) method (Taylor et al., 2015). The refractive indices of the BC core and

coatings in the derivation of $D_p$ were 2.26-1.26i (Moteki and Kondo, 2008) and 1.48-0i (Taylor et al., 2015), respectively, at a wavelength of 1064 nm. Aquadag (Acheson Inc., USA) and polystyrene latex sphere (PSL, Nanosphere Size Standards, Duke Scientific Corp., USA) aerosols were used to calibrate the incandescence and scattering signals of SP2. Because of the higher incandescence sensitivity of ambient BC than Aquadag (Moteki and Kondo, 2010), the calibration was scaled following the recommendation from (Laborde et al., 2012). The data acquisition mode of SP2 was set to record 1 particle every 2 particles

to guarantee the high resolution of the vertical profile. More specific details regarding calibration and data processing can be found in our previous literature (Liu et al., 2020b). In general, the size distribution of the BC core ($D_c$) and coating thickness ($D_p/D_c$) could be determined through SP2. The mass median diameter (MMD) was used to quantify the size distribution of the BC core and calculated from the $D_c$ distribution below and above which the BC mass was equal (Liu et al., 2019a). The $D_p/D_c$ of bulk BC aerosols was calculated using the $D_p/D_c$ of every single BC particle in a certain time window:

$$\frac{D_p}{D_c} = \sqrt[3]{\frac{\sum_i D_{p,i}^3}{\sum_i D_{c,i}^3}} \tag{1}$$

The hygroscopicity of BC-containing particles was calculated based on the volume-weighted Zdanovskii-Stokes-Robinson (ZSR) rule (Stokes and Robinson, 1966) by assuming that the hygroscopicity parameters of BC and coatings were 0 and 0.3, respectively (Pringle et al., 2010). A detailed calculation of the hygroscopicity can be found in (Hu et al., 2021a).

### 2.2.2 Optical properties

The absorption efficiency ($b_{abs}$) at a wavelength of 880 nm was directly measured by an Aethalometer (AE33, Magee Scientific Corp.) The scattering efficiency ($b_{sca}$) at a wavelength of 870 nm was measured by a photoacoustic extinctiometer (PAX, Droplet Measurement Technologies). A PM$_{2.5}$ cyclone with a supporting pump was placed before AE33 and PAX; thus, the measured babs and b$_{sca}$ are characteristics of PM$_{2.5}$. The single scattering albedo (SSA) at a wavelength of 870 nm was calculated as follows:

$$SSA = \frac{b_{sca,870\,nm}}{b_{sca,870\,nm} + b_{abs,870\,nm}} \tag{2}$$

The babs at a wavelength of 880 nm was transferred to babs at a wavelength of 870 nm through Equation 3:



$$b_{abs,870\,nm} = b_{abs,880\,nm} * \left(\frac{870}{880}\right)^{-AAE} \tag{3}$$

The absorption Ångström exponent (AAE) is assumed to be 1 in the calculation since brown carbon has little effect at long
wavelengths.

The absorption enhancement ($E_{abs}$) of BC aerosols at a wavelength of 550 nm is calculated based on the information of $D_c$ and
$D_p$ from SP2's measurement by using the Mie core-shell approach. The refractive indices used in the calculation are 1.95+0.79i
for the BC core (Bond and Bergstrom, 2006) and 1.50+0i for the coatings (Liu et al., 2015). The core-shell assumption may
overestimate the $E_{abs}$ (Liu et al., 2017;Wu et al., 2018;Liu et al., 2019c) because of the complex morphology of BC (Wang et
al., 2017;Hu et al., 2021b). Liu et al. (2020b) estimated that such overestimates could be ~17% in summer in Beijing.

## 3. Results

### 3.1 Overview of the observation

The time series of the major observation parameters during the campaign are illustrated in Fig. 1. The height of the moveable
container was also shown in Fig. 1a. Thus, the time series in Fig. 1 also reflects the vertical information of the pollutants. The
average concentration of BC was 0.34 μg/m$^3$, and the average mixing ratio of $O_3$ was 33 ppb. The relative humidity and
temperature were high during the summer, with average values of 67% and 27 ℃, respectively. In general, the air condition is
relatively clean during the observation period due to the frequent rainy event, except for a heavy $O_3$ pollution event that
occurred on the 25th of June with a maximum hourly mixing ratio of 124 ppb.

The mass size distribution of the BC core followed a lognormal distribution with a mass median diameter (MMD) of 173 nm
and a geometric standard deviation (GSD) of 1.57, as shown in Fig. S1, consistent with the MMD value (171 nm) observed in
Beijing in the summer of 2018 (Liu et al., 2020b). The MMD value remained stable between 160-190 nm. The coating
thickness ($D_p/D_c$) of the BC bulk varied largely from 1.1 to 1.6, with an average value of 1.33. The coating thickness of BC
could be as high as 1.4-1.5 even in the relatively clean conditions during July 1st to the 5th. Figure S2 exhibits the vertical
meteorological conditions. The wind direction from July 1st to 5th was dominated by the south wind. The wind was mainly
from the north during other periods. The transportation of aged BC from the south may lead to an increase in coating thickness
even under clean conditions. Such an increase in $D_p/D_c$ was also found from the 10th of July to the 12th, during which the
wind also came from the south. Thus, the coating thickness may be highly dependent on the air mass. Zhang et al. (2018)
reported that the transportation of aged BC from regional sources outside Beijing in winter and BC from transportation could
account for 63% of total BC aerosols. Such transportation may also be efficient in summer. It is also noted that the $D_p/D_c$ value
could increase quickly from 1.1 to 1.4 during the ozone pollution day (5th July), and the severe photochemical process may
also be attributed to the increase in the BC coating thickness.





### 3.2 Classification of the vertical profile

Fig. S3-Fig. S21 exhibit all measured vertical profiles of different parameters during the observation. Based on the vertical
profiles of BC concentration, the vertical profiles were classified into four categories, and Fig. 2 shows the typical profiles of
the four categories.

#### 3.2.1 Uniform type

The vertical profile at 12:00 on 30th June was selected to represent the uniform type. As shown in Fig. 2a, the concentration
of BC varied little from the surface to 240 m in height in this type. In addition to BC concentration, the microphysical properties
(such as MMD and $D_p/D_c$) and optical properties (such as $b_{abs}$, $b_{sca}$ and SSA) also varied little. The uniform aerosol vertical
distribution indicates sufficient mixing in the lower boundary layer. In this case, the temperature of the surface was ~4℃
higher than that at 240 m height. Such a large temperature difference would promote the vertical motion of the air mass and
contribute to vertical mixing. Although vertical mixing is efficient in this case, the ozone concentration was found to be lower
in the near-surface area, and the vertical profile exhibits a uniform distribution above ~50 m height. The presence of substantial
$NO_x$ emissions from traffic sectors would decrease the concentration of ozone through the titration reaction between ozone
and NO, leading to the decreasing trend of ozone concentration with height.

#### 3.2.2 Gradual decrease type

The vertical profile at 18:00 on 2nd July was selected to represent the gradual decrease type. In this type, the BC concentration
gradually decreases with height. Such a vertical profile is mainly attributed to the stable boundary layer as well as weak
convection and turbulence. The continuous emission of BC from the surface leads to a higher concentration near the surface.
Fresh BC emissions would also lead to a decrease in $D_p/D_c$ near the surface, resulting in an increasing trend of $D_p/D_c$ with
height. The aging process is efficient in increasing the coating thickness but not the size of the BC core. The MMD value
remained stable at all heights in this case. The size of the BC core is reported to vary among different emission sources
(Schwarz et al., 2008;Pan et al., 2017). The nearly stable MMD value in this case may suggest similar sources of BC in the
lower boundary layer, but the BC in the upper heights had a longer aging time (larger $D_p/D_c$). The concentration of $O_3$ decreased
with increasing height at a sharper slope than that in the uniform case.

#### 3.2.3 Sharp decrease type

The vertical profile at 23:00 on 27th June was selected to represent the sharp decrease type. This type is also referred to as the
'stratified type' in other studies. The concentration of pollutants decreased sharply at a certain height but remained uniform or
slightly decreased with height above a certain height. Profiles of this type mainly appeared in the early morning and midnight.
The $D_p/D_c$ was larger and $\triangle BC/\triangle CO$ was smaller at the upper height, indicating more aged BC at the upper height.

The vertical profiles at 22:00 on 27th June and 23:00 on 27th June are simultaneously shown in Fig. S7. As shown, the vertical profile at 22:00 exhibited a gradual decreasing trend. However, the profile changed into a sharp decrease type 1 hour later. Comparing the vertical profiles at these two hours, the BC concentration was similar above 200 m, but the concentration

significantly increased below 200 m at 23:00. The $D_p/D_c$ was also similar above 200 m and significantly decreased below 200 m at 23:00. The BC concentration and $D_p/D_c$ characteristics in these two profiles suggest that freshly emitted BC was trapped at lower heights (below 200 m) at 23:00. The top of the night boundary height may be 160~200 m at 23:00 in this case. Such a stratified vertical distribution is also found in previous studies (Wang et al., 2018a;Guimaraes et al., 2019;Lei et al., 2021). Lei et al. (2021) found that the boundary layer heights measured by ceilometer were relatively low (~110-250 m) in the

stratified type. Wang et al. (2018a) used the sudden concentration transition point to denote the top of the boundary layer. Thus, it is plausible to distinguish the night boundary layer and residual layer through the sudden concentration decrease point, and the profiles of the sharp decrease type were used to study the pollutant difference between the night boundary layer and residual layer in the following section.

### 3.2.4 Sudden increase type

The vertical profile at 11:00 on the 11th of July was selected to represent the sudden increase type. As shown in Fig. 2d. The BC mass concentration rapidly increased from 0.4 to 1.2 μg/m3 at 80 m above the ground and quickly decreased to the background level above 90 m. The concentration of $NO_x$ and the mixing state of BC also exhibited a sudden change. The MMD and $D_p/D_c$ experienced a sudden decrease, while the $NO_x$ concentration increased sharply. The characteristics of MMD, $D_p/D_c$ and $NO_x$ indicate that the sudden increase in this case may come from fresh emissions from traffic sources, since BC

from traffic sources generally has a small BC core and thin coatings (Liu et al., 2017;Holder et al., 2014), and the $NO_x$ emissions from traffic sources are also substantial.

Among the 7 sudden increase cases, 4 sudden increases occurred at ground level, and 3 sudden increases were found in midair. In addition to the emissions from traffic sources (characterized by the low MMD, $D_p/D_c$ and simultaneous increase in $NO_x$), the emissions from other sources could also lead to a sudden increase. The sudden increase that occurred at 11:00 on the 9th

of July, as shown in Fig. S9, was accompanied by an increase in MMD, a decrease in $D_p/D_c$ and no significant variation in $NO_x$ concentration. The sudden increase in this case may be influenced by the emission from solid fuel burning with a larger MMD.

### 3.2.5 Other profiles in special cases

Fig. S6 shows the profiles on the ozone pollution day (maximum hourly ozone mixing ratio reached 124 ppb), the vertical

profiles of BC, including its mixing state, and ozone presented in a uniform type. Since the formation of ozone is always accompanied by high temperature and radiation leading to highly sufficient mixing in the boundary layer, the uniform vertical distribution of pollutants may be the common situation on ozone pollution days.





Another special profile is the profile (19:00 1st July, shown in Fig. S11) after a heavy rainy event. The moveable container started rising at 18:00 and reached a height of 240 m at 18:30. When the container stayed at the tower, heavy rain happed. The
rainy event lasted for nearly 80 minutes, and the container started moving to the ground as soon as the rain ended. Thus, the vertical profile between 18:00 on 1st July and 19:00 on 1st July could be compared to analyze the effect of wet scavenging. As shown, these two vertical profiles both belong to the uniform type, indicating that the wet scavenging efficiency may be vertically similar in the lower boundary layer or that there may be quick vertical mixing during the daytime in summer. It is noted that the MMD decreased largely from 190 nm to 170 nm within 1 hour. The bulk $D_p/D_c$ value remained invariant, but
the $D_p/D_c$ for $D_c$ = 180 nm decreased from 1.32 to 1.25. The variation in MMD and $D_p/D_c$ suggests that wet scavenging may prefer to remove large BC cores with thicker coatings. The preference of wet scavenging for BC with large sizes has also been reported in previous research (Liu et al., 2020b , Wang et al., 2015 , Taylor et al., 2014). As also exhibited in Fig. 1c, the MMD tended to decrease during other rainy events when the container was at ground level, but the $D_p/D_c$ showed less variation.

### 3.3 Diurnal variation in the vertical profile

Fig. 3 shows the number fractions of vertical profile types at different times during the day. The uniform type is the dominant type, accounting for ~ 80% during the daytime (8:00–18:00). The number fraction of the uniform type started dropping at 19:00 and only accounted for 14% at 23:00 and rose back after sunrise. In contrast, the number fraction of the gradual decrease type started rising at night and dropped back ~10% during the daytime. The variation in the boundary layer leads to such vertical profile type changes. Due to the much higher heat capacity of the surface than that of air, the temperature of the surface
responds to radiation at a much faster rate, leading to the boundary layer's thermal structure and stability change, especially in summer with strong radiation. Thus, the vertical motion and mixing during the daytime is more severe than that at night, and a uniform profile is the most common case during the daytime. In fact, the several gradual decrease cases observed during the daytime generally occurred on cloudy days when the radiation was relatively weak. The sudden increase type always occurred during the daytime, partially caused by the more frequent human activity at the time, resulting in occasional plume
transportation to the measurement site. The sharp decrease type, indicating the low night boundary layer height, occurred five times during the whole observation and was found only at midnight (23:00) or the early morning (6:00-7:00).

In addition to the BC concentration, the average vertical profiles of other parameters at 7:00, 12:00, 18:00 and 23:00 are counted to further study the diurnal variation, as illustrated in Fig. 4 (the averaged profiles with error bars could be found in Fig. S26). The average value and the shape of vertical profiles both have clear diurnal changes.
For the average value, the mass concentration of BC is higher and $D_p/D_c$ is lower at night and in the morning than during the daytime. The changes are mainly attributed to the fresh BC emissions and low night time boundary layer height. Fresh BC tends to accumulate in the night boundary layer. The development of the boundary layer and photochemical aging with the condensation process during the daytime would result in a decrease in the concentration and an increase in $D_p/D_c$. Since $O_3$ is the product of photochemical reactions, the concentration of $O_3$ is higher at noon and significantly decreases at night. The
concentration of $NO_x$ exhibits the opposite variation trend with $O_3$.



For the shape of the vertical profile, the average profiles of all parameters during the daytime (12:00 and 18:00) generally follow a uniform distribution except $O_3$. The $O_3$ concentration exhibits a slight decrease at the surface level at 12:00, and the degree of decrease extends at 18:00.

Compared with the uniform shape during the daytime, the vertical gradients of the average profiles at night and in the morning (7:00 and 23:00) are larger. For BC aerosols, the average concentration decreases with height at a rate of -0.075 µg/m3 per 100 m at 23:00. The $D_p/D_c$ is nearly uniform below 160 m and slightly decreases with height above 160 m. The transition at 160 m is mainly affected by the two sharp decrease profiles at 23:00, which both occurred at ~160 m height. Excluding the two sharp decrease types, the average $D_p/D_c$ shows little variation with height. The MMD also remained nearly invariable at 0-240 m. The variation tendency of the BC concentration, $D_p/D_c$ and MMD at 7:00 was similar to that at 23:00, but the vertical gradient at 7:00 was lower. The vertical profile of $NO_2$ shows a very similar variation trend with that of BC concentration, partially caused by the similar emission sources of these two pollutants in the urban area. The concentration of NO was near zero at 12:00, 18:00 and 23:00 and significantly increased at 7:00 due to the low $O_3$ concentration at the time as well as the large emissions from the morning commute. The vertical profile of NO at 7:00 shows a decreasing trend with increasing height. The average vertical profile of $O_3$ increased with increasing height at 7:00 and 23:00.

### 3.4 Pollutant properties in the residual layer

The BC properties, $O_3$ and $NO_x$ at the 240 m height and ground are counted for the 5 sharp decrease profiles to study the pollutant difference between the ground and residual layers. For BC aerosols, the concentration is much higher on the ground than in the residual layer. The BC concentration on the ground was 84.6% higher than that in the residual layer at 23:00 on the 8th of July. BC in the residual layer generally had more coatings, indicating a higher aging degree. The maximum difference in $D_p/D_c$ was found at 23:00 on 27th June, and the $D_p/D_c$ in the residual layer was 8.4% higher than that on the ground. The difference in BC concentration and $D_p/D_c$ in the two layers could be explained by the accumulation of fresh BC in the night boundary layer. In addition, as exhibited in Fig. S22, the number frequency of another BC type with an apparently higher $D_p/D_c$, denoted by the red circle, increased at 240 m height. The transportation of aged BC from other areas in the residual layer may also lead to an increase in $D_p/D_c$. The MMD values were relatively stable (~175-185 nm) on the ground, while the MMD in the residual layer was more variable, ranging between 200 nm and 170 nm. Since the BC core is inertial in the atmosphere, the size distribution of the BC core is mainly affected by the emission sources and has been used as an index for source apportionment (Wu et al., 2017). The stable MMD may indicate stable emission sources in the boundary layer, and the MMD value on the ground is close to the reported MMD value observed at the urban site, which is mainly influenced by traffic emissions (Liu et al., 2020b;Wu et al., 2021b). The variable MMD value in the residual layer may indicate more complicated sources. Other physical properties, such as absorption ability and hygroscopicity, are calculated and exhibited in Fig. 5e-f. Due to the thicker coating, BC in the residual layer generally has a higher absorption ability and hygroscopicity. At 23:00 on the 27th, the absorption enhancement ($E_{abs}$) in the residual layer is 8.6% higher than that on the ground, and the critical supersaturation point in the residual layer is 25.0% lower than that on the ground.



The $O_3$ concentration showed an opposite and much more significant difference between the ground and residual layer
compared with the concentration of BC. In the case at 23:00 27th, the concentration of $O_3$ at 240 m height was 226.6% higher
than that on the ground, since the higher $NO_x$ concentration on the ground would decrease the $O_3$ in the night boundary layer
at night.

Previous studies found that the higher $O_3$ concentration in the residual layer could increase the $O_3$ concentration on the ground
in the early morning with the development of the mixing layer and entrainment of $O_3$ from the residual layer (Xu et al., 2018;Hu
et al., 2018). The transportation of $O_3$ from the residual layer to the ground was also observed in this study. As shown in Fig.
6, the $O_3$ concentration was nearly zero before 6:00. Two vertical measurements were conducted during 6:00-8:00, and a much
higher $O_3$ concentration was observed in the upper height. The $O_3$ concentration significantly increased at 8:00 when the
moveable container returned to the ground compared with the $O_3$ concentration at 6:00, but the concentration stayed stable
between 8:00 and 10:00 and rose again after 10:00. The stable concentration of $O_3$ between 8:00-10:00 means that the
photochemical reaction may not be very efficient before 10:00 in this case. The increase in the $O_3$ concentration on the ground
between 6:00 and 8:00 may not be mainly attributed to photochemical reactions but to transportation from the upper height
with high $O_3$ concentrations. Such development of the boundary layer and vertical mixing was caused by the quicker heating
of the ground (Fig. 6e) and would decrease the $O_3$ concentration in the upper heights in turn. As denoted by the gray dashed
line in Fig. 6d, the $O_3$ concentration remained invariable at the beginning and then started to decrease when the container was
located at 240 m height during 6:30-7:15.

A similar variation tendency was also found for the $D_p/D_c$ value, as shown in Fig. 6c. The aged BC in the residual layer could
also increase the $D_p/D_c$ on the ground in the early morning through entrainment and vertical mixing, even though the fresh BC
emissions from traffic sources are substantial at the time. Another typical case of the increase in $D_p/D_c$ and $O_3$ through vertical
mixing occurred on 11th July. A similar increase in the early morning and stable condition at the later hours was found as
exhibited in Fig. S23. In the cases with enough photochemical reactions, stable conditions after the early morning increase
may not be found, and the concentrations of $O_3$ and $D_p/D_c$ continuously increase from the early morning. However, it is
reasonable to infer that vertical mixing could also contribute to the increase in $O_3$ and $D_p/D_c$ in the early morning, since $O_3$ and
$D_p/D_c$ were higher in the residual layer in most cases. From the profiles of $O_3$ and $D_p/D_c$ on July 13[th] (Fig. S19), the pollutants
from the residual layer firstly influence the upper boundary layer denoted by the large difference of $O_3$ and $D_p/D_c$ between 240
m and the surface at 6:00. Then, with the development of vertical mixing due to the sun rise, the pollutants at higher height
reached to the surface. The $O_3$ and $D_p/D_c$ at the surface increased and vertical profiles became uniform at 7:00. Such
phenomenon could be also reflected on the averaged profiles (Fig. 4), since the averaged value of $O_3$ and $D_p/D_c$ were higher at
240 m at 7:00.

The distinguished pollutants in the residual layer could also influence the chemical composition of particulate matter in the
boundary layer (Lei et al., 2021). Due to the limitation of the observation height, the pollutant conditions at the top of the
boundary during the daytime layer could not be detected. Liu et al. (2019b) reported rapid BC aging at the top of the boundary
layer in summer due to intensive actinic flux. The $D_p/D_c$ above the boundary layer is reported to be higher than that on the





ground under clean conditions (Ding et al., 2019). Thus, the Dp/Dc increase in the boundary layer due to vertical transportation may not only occur in the early morning, but entrainment from the lower troposphere may also lead to a $D_p/D_c$ increase in the
daytime.

**3.5 Optical properties of black carbon**

The optical properties of BC aerosols are of most concern due to BC's strong absorption ability. Fig. S24 exhibits the relationship between BC concentration and $b_{abs}$ during the whole observation period. In general, $b_{abs}$ increased with BC concentration. For the same BC concentration, $b_{abs}$ increased with $D_p/D_c$ through the so-called "lensing effect". The "lensing
effect" also appeared in the vertical distribution. As shown in Fig. 4, the vertical distribution of $E_{abs}$ shows a similar vertical variation in $D_p/D_c$ with a uniform distribution during the daytime and a slight increase at 150-240 m at night. The $E_{abs}$ at 240 m height is ~1-2% higher than that on the ground on average at night. The vertical difference in $E_{abs}$ caused by thicker coatings is significantly lower than that of the BC concentration, as shown in Fig. 7. The vertical difference in BC concentration reached -40% - 40% but was only -5%-5% for $D_p/D_c$. Thus, the vertical distribution of $b_{abs}$ is mainly determined by the variation in BC
concentration, as proven by the similar average vertical profiles of BC concentration and $b_{abs}$. For the averaged profiles at 23:00 (the profiles with most significant vertical difference), $b_{abs}$ at 240 m is 29.3% lower than that at ground. Such decrease could reach 30.3%, if the same coating thickness was used at all heights. In other words, the increase in absorption enhancement at upper height due to coatings could slightly decrease the vertical difference of $b_{abs}$, but couldn't compensate for the decrease of $b_{abs}$ at upper height due to the much more significant decrease of BC concentration.

Single scattering albedo (SSA) is another key optical parameter for aerosols since the influence of aerosols on surface temperature could change from cooling to heating when the SSA is lower than the critical point of 0.90-0.93 (Hansen et al., 1997;Cook and Highwood, 2004). Although the "lensing effect" could increase the absorption ability for BC-containing aerosols, the scattering of BC-containing aerosols simultaneously increases with $D_p/D_c$, leading to an increase in SSA, as shown in Fig. S25. For the bulk aerosols (including BC and non-BC aerosols) in the ambient atmosphere, the SSA also
increases with increasing $D_p/D_c$ (Fig. S24b). The increase in SSA for bulk aerosols with $D_p/D_c$ may be influenced by the higher SSA for BC-containing particles with higher $D_p/D_c$. Another reason is that the BC fraction in bulk aerosols may be lower in the airmass with a higher $D_p/D_c$, since $D_p/D_c$ is an indicator for secondary aging processes and BC is the product of primary emissions. The vertical distribution of SSA exhibited a uniform type during the daytime. At night time, $b_{sca}$ and $b_{abs}$ generally decrease with increasing height, but $b_{abs}$ decreases at a quicker rate, resulting in an increase in SSA with height at night. Such
an SSA variation trend has also been reported in previous vertical measurements (Li et al., 2022b).

**4 Conclusion and remarks**

Continuous vertical measurements were conducted in the summer of 2022 to study the vertical distribution of BC and its mixing state in the boundary layer. Based on the vertical distribution of BC concentrations, the vertical profiles could be





classified into four categories: uniform, gradual decrease, sharp decrease and sudden increase. The uniform type indicates

sufficient vertical mixing in the boundary layer and could account for ~80% of the total profiles during the daytime. In the uniform type, all pollutant concentrations except $O_3$ are nearly the same along 0-240 m. The gradual decrease and sharp decrease types suggest stable vertical conditions and could account for ~40-90% of the total profiles at midnight and early morning. The concentrations of pollutants were much higher on the ground due to substantial emissions and were difficult to transport to higher altitudes due to the stable boundary layer structure. The sudden increase type is caused by the fresh plume

emission, which is always accompanied by a significant increase in BC concentration at a narrow height range and a decrease in $D_p/D_c$. The vertical variation of $O_3$ is different from other pollutants, and the $O_3$ concentration generally increases with height even in the uniform type for other pollutants due to the large emission of $NO_x$ on the ground.

The mass median diameter (MMD) and shell/core ratio ($D_p/D_c$) were used to represent the size distribution of the BC core and BC coating thickness. The MMD and $D_p/D_c$ generally followed a uniform vertical distribution in the boundary layer. Under

some stable conditions, such as a night boundary layer or cloudy days, $D_p/D_c$ increased with height. However, the vertical difference in BC concentration is much more significant than that in $D_p/D_c$ under stable conditions. For the average profiles at 23:00, the BC concentration at 240 m height could be 34.6% lower than that on the ground, but the $D_p/D_c$ at 240 m height was only ~2% higher than that on the ground, leading to 1-2% absorption enhancement ($E_{abs}$) difference. Although the coatings could increase the absorption, they increase the absorption in similar degree in the lower boundary layer. The vertical variation

in BC concentration plays a more important role in the vertical difference of BC's absorption ability.

Vertical profiles with BC concentrations experiencing a sudden decrease at a certain height were classified as the sharp decrease type and were used to study the pollutant difference in the residual layer and boundary layer. Compared with the ground, the residual layer generally has BC with a higher $D_p/D_c$, lower concentration and more variable MMD. Such a vertical difference is much more significant than the vertical difference in the boundary layer. In special cases, the $D_p/D_c$ in the residual

layer could be 8.4% higher than that on the ground, which could lead to an 8.6% absorption increase and 25.0% critical supersaturation decrease. The pollutants with distinct properties in the residual layer would influence the pollutants in the boundary layer in the early morning through entrainment and vertical mixing with the development of the boundary layer. It's observed aged BC and high concentration $O_3$ in the residual layer could lead to a quick increase in $D_p/D_c$ and $O_3$ concentrations on the ground in the early morning (6:00–8:00). Since the height of the boundary layer is much higher in summer than in

winter, only limited sharp decrease type profiles were observed in this study. In winter, BC generally experiences a quicker aging process and has thicker coatings. Such highly aged BC may stay in the residual layer at night. With more fresh BC emission sources and a lower night boundary layer, the vertical difference in BC concentration and its microphysical properties between the ground and residual layers may be even larger and play a more important role in boundary layer development in the early morning in winter. In this study, continuous vertical profiles of BC and its mixing state as well as the optical properties

were reported, and obvious diurnal changes in vertical profiles were found due to the strong radiation in summer. In addition, the pollutants in the residual layer could increase the $O_3$ concentration and $D_p/D_c$ value of the surface level in the early morning.



These findings provide more knowledge about the vertical distribution of BC in the boundary layer and will be helpful for estimating the regional climatic and environmental effects of BC.

**Author contributions.**

Hang Liu: Conceptualization, Methodology, Software. Xiaole Pan: Conceptualization. Shandong Lei: Investigation. Yuting ZHANG: Investigation. Aodong Du: Investigation. Weijie Yao: Investigation. Tao Wang: Visualization. Jinyuan XIN: Resources. Jie Li: Visualization. Yele Sun: Resources. Junji CAO: Resources. Zifa Wang: Resources, Supervision.

**Data availability**

To request the data given in this study, please contact Dr. Hang Liu at the Institute of Atmospheric Physics, Chinese Academy of Sciences, via email (liuhang@mail.iap.ac.cn).

**Competing interests**

The authors declare that they have no conflict of interest.

**Acknowledgment**

This work was supported by the National Natural Science Foundation of China (No. 92044301, No. 42177092, No. 41877314) , the National Key Research and Development Program of China (2022YFC3701000, Task 4), and the China Postdoctoral Science Foundation (No 2022M713094). We thank the Public Technology Service Centre, Institute of Atmospheric Physics, Chinese Academy of Sciences for the technique support during the measurements.

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





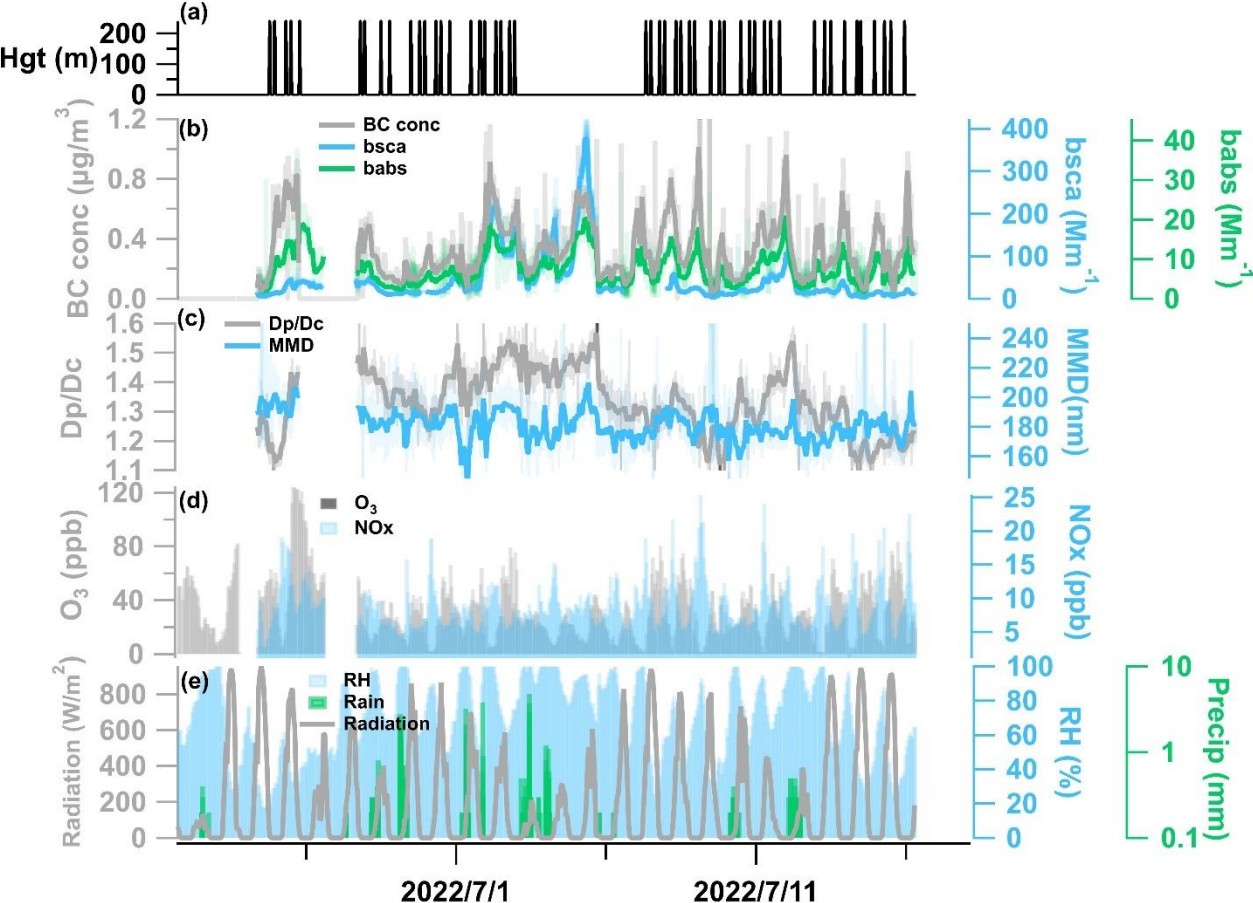

**Figure. 1 Time series of the major observed parameters during the observation; the data were processed to a 1-hour resolution. (a) The height of the moveable container. (b) Mass concentration of black carbon and absorption ($b_{abs}$) and scattering efficient ($b_{sca}$), the light-coloured lines denote the data with 1-min resolution. (c) The coating thickness (D$_p$/D$_c$) and mass median diameter (MMD) of BC, the light-coloured lines denote the data with 1-min resolution. (d) The mixing ratio of O$_3$ and NO$_x$. (e) The meteorological conditions during the observation period.**







**Figure. 2 Typical profiles of the pollutants and parameters of the four types. The four types were classified based on the vertical profiles of BC mass concentration. The exhibited parameters from left to right are BC mass concentration, aerosol absorption efficient ($b_{abs}$), aerosol scattering efficient ($b_{sca}$), single scattering albedo (SSA), BC coating thickness ($D_p/D_c$), mass median diameter of BC core (MMD), CO mixing ratio, $\triangle BC/\triangle CO$, mixing ratio of $O_3$ and $NO_x$, as well as relative humidity (RH) and temperature (T).**



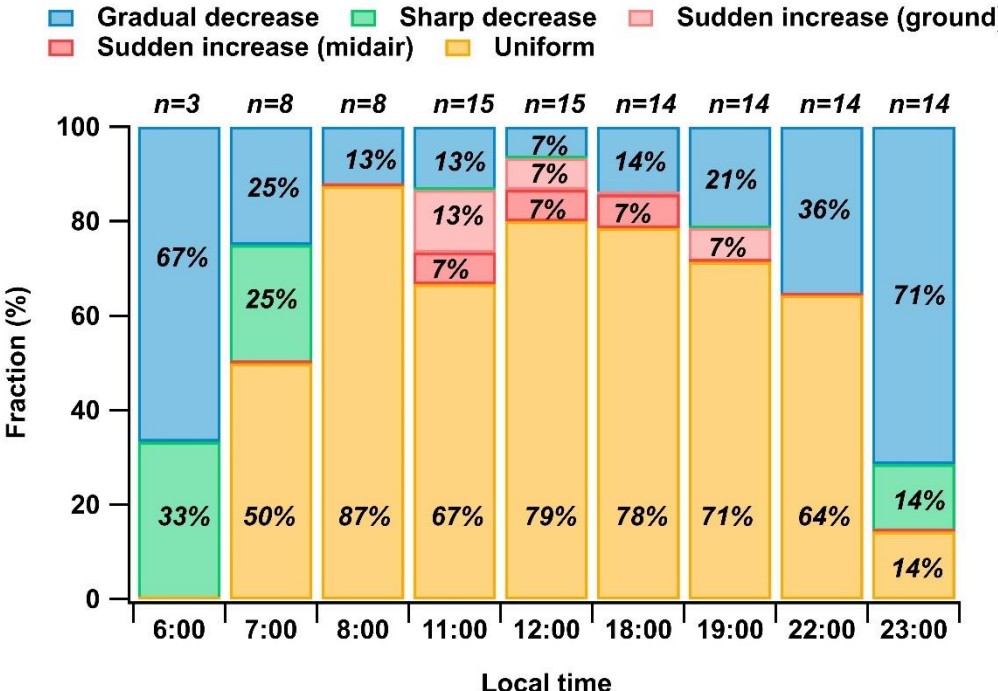

**Figure. 3** Number fraction of the different vertical profile types at various times. The sudden decrease type was further classified into two types based on the sudden increase location. *n* above each bars denotes the total measured profile numbers at the time.



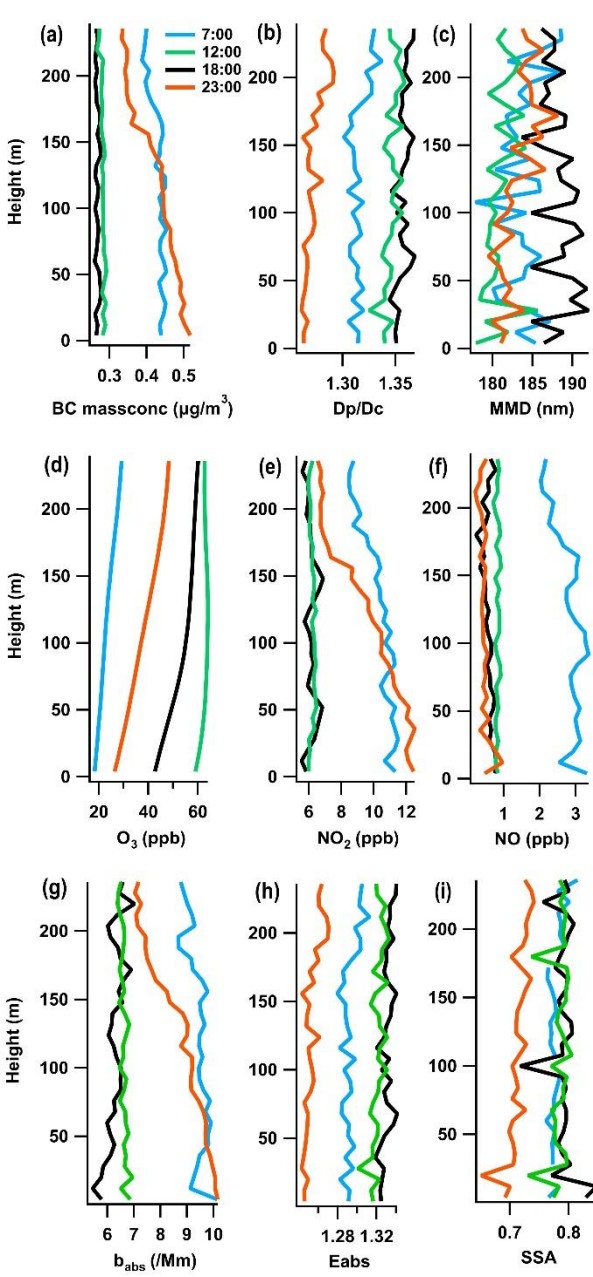


**Figure. 4 The diurnal variation of the average profiles of BC properties including (a) BC, (b) D$_p$/D$_c$, (c)MMD. The average profiles of (b) O$_3$ and its precursors (e) NO$_2$ and (f) NO. The averaged profiles of aerosol optical properties including (g) absorption efficient, (h) absorption enhancement of BC due to coatings, (i) single scattering albedo.**





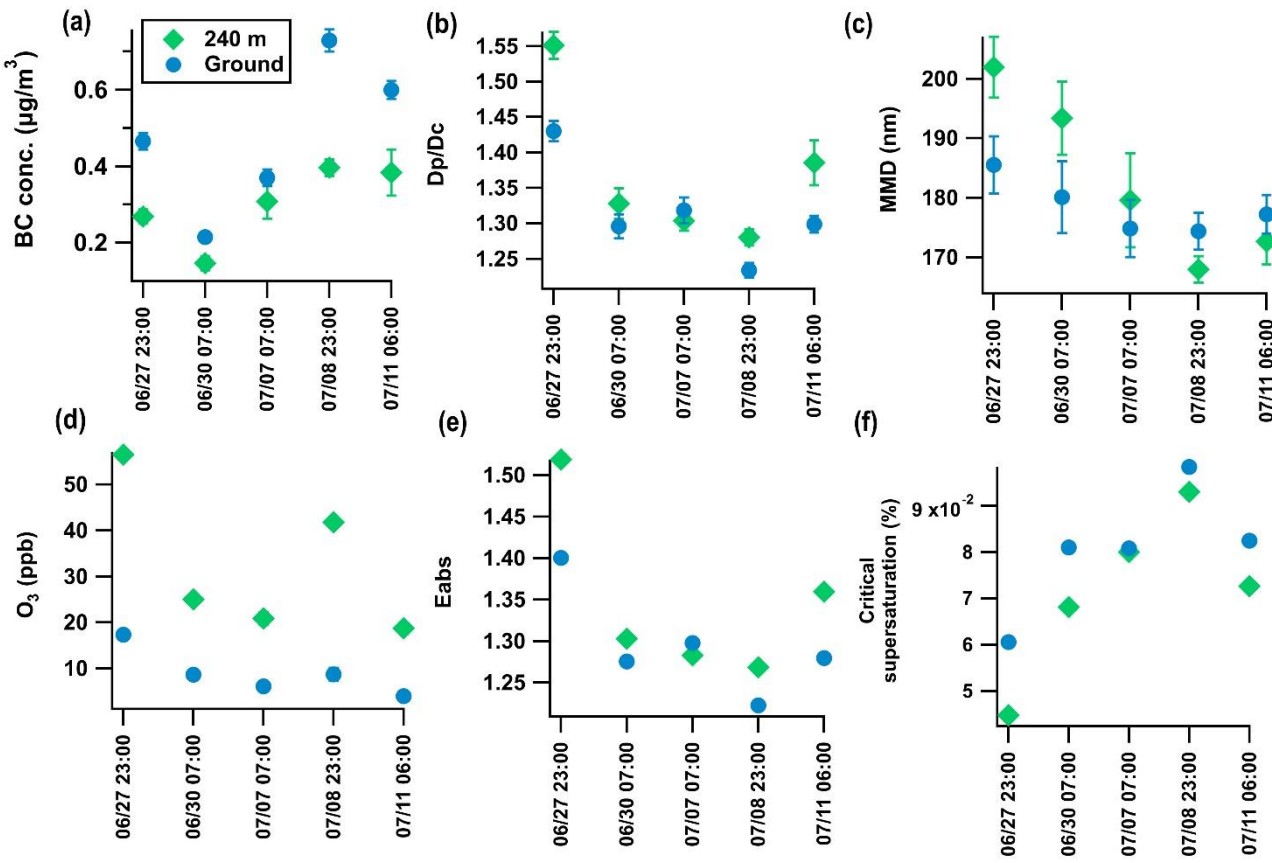


**Figure. 5 Comparison between the surface and 240 m height in the sharp decrease type of major pollutant concentrations (BC, O₃) and BC micro physical characteristics including coating thickness (Dₚ/Dᴄ), mass median diameter of BC core (MMD), the absorption enhancement of BC due to coatings (Eₐᵦₛ) and critical supersaturation point of BC.**





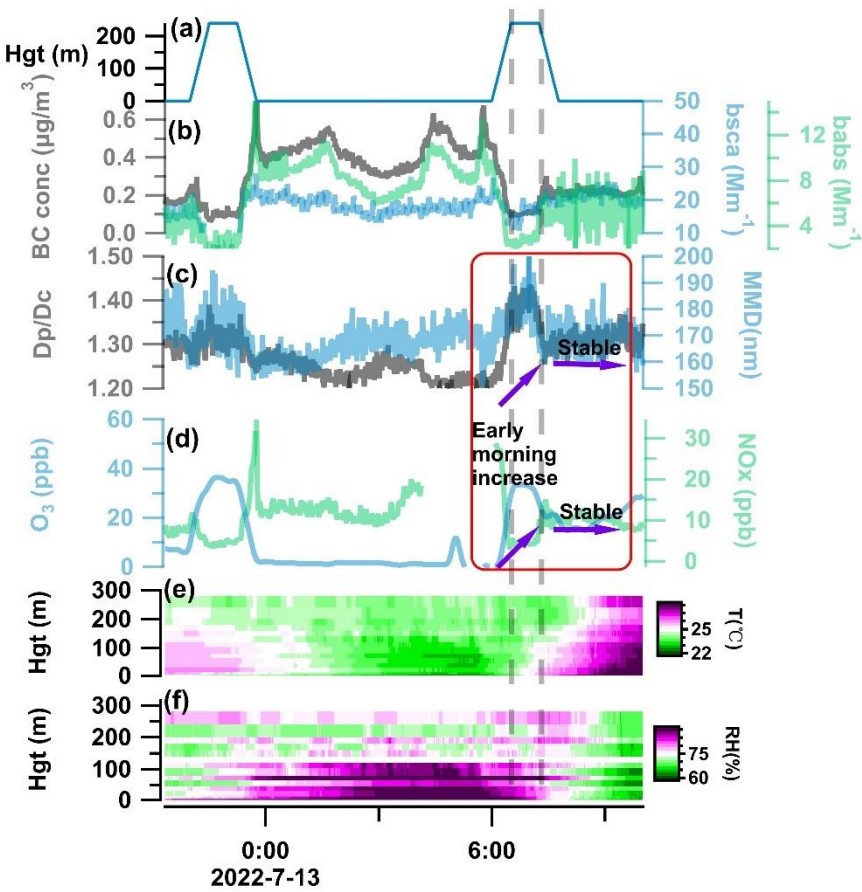

Figure. 6 A typical case of vertical mixing from the residual layer leading to an increase in O₃ and $D_p/D_c$ in the early morning. (a) The height of moveable container. (b)-(d) Time series (1-min resolution) of BC mass concentration, $b_{abs}$, $b_{sca}$, $D_p/D_c$, MMD, O₃ and NOₓ. (e) 2D (height and time) plot of temperature, the colours denote the quantity of the temperature. (f) The same as e, but for relative humidity.





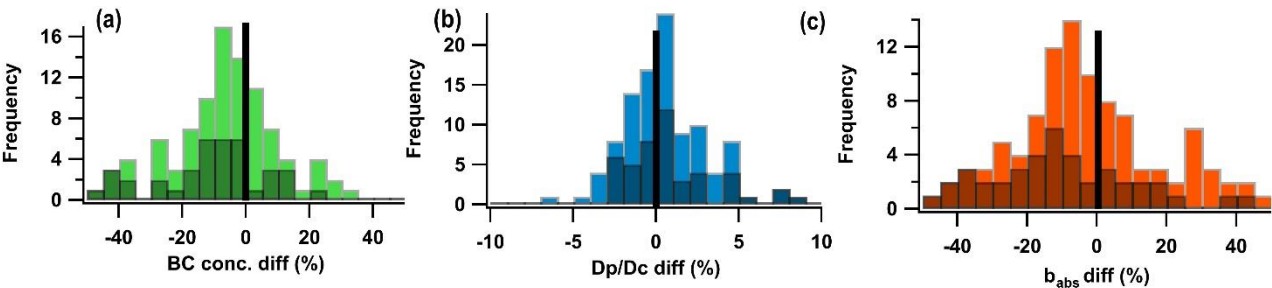

**Figure. 7 Histogram of the number frequency of the relative differences between 240 m and the ground for a) BC ,(b)**
**$D_p/D_c$ and (c) $b_{abs}$. The shaded black area denotes the number frequency at night and morning (6:00,7:00, 8:00, 22:00,**
**23:00), the black line denotes no difference between 240m and the ground.**