# Peer review of "Vertical distribution of black carbon and its mixing state in urban boundary layer in summer"

_Atmospheric Chemistry and Physics, 2022_

## Referee Comment (RC2)

The authors present vertical (0-240 m) measurements of BC and its mixing state during summertime in urban areas in China, focusing on analysis and interpretation of four types of vertical profiles in the boundary layer. The discussion and conclusion could benefit from some quantitative analysis, e.g., the diurnal variation in the vertical profiles. This work is interesting and suitable for publication within the scope of ACP. Please see detailed comments below.

General Comments:

The authors can give more discussion on the significance of the work. In earlier studies, what had been left open, and what the present paper now to address? How their finding are relevant to atmospheric chemistry and radiative transfer? What the novel insights of their work are?

Specific comments:

(1) Abstract. Please indicate where the measurements are taken in the abstract and the date range of the measurements.

(2) SP2 data analysis: The minimum core size (from LII) and minimum total particle size (from scattering) differ, with the latter greater than the former. Have the authors filtered to account for any mismatch in the SP2 size for incandescence vs scattering, which can bias results? What are the limits on the Dp? When calculating BC concentration calculation, have the authors considered the SP2 limitations?

(3) Page 5/Line 149-151: The authors are trying to add some discussions on the causes of BC coating growth during ozone pollution day. However, there is no sufficient evidence to support. The authors can support their explanation by previous studies, e.g. https://doi.org/10.1021/acs.est.2c00090.

(4) Page 6/Line 165-166: The author can give NO data rather than NOx to demonstrate titration reaction.

(5) Page 6/Line 173-174: The author can discuss the emission sources of the measured BC based on MMD.

(6) Page 6/Line 175-176: O3 decreased with increase height? I think that should be "O3 increased with ......".

(7) Page 7/Line 296-308: The measurement found higher O3 and Dp/Dc in the residual layer. The authors can give more discussion on BC aging in the residual layer. How the finding is relevant to atmospheric chemistry and radiative transfer?

(8) Conclusions: The authors need to clear the novel insights of their work and the significance of their findings.

---

## Author Comment (AC1)

**Reply to the comments of anonymous reviewer #2 on manuscript Entitled " Vertical distribution of black carbon and its mixing state in urban boundary layer in summer"**

We sincerely appreciated the comments from the reviewer. Many advices from the reviewer are really helpful in the improving of this paper. We have carefully considered the comments and revised the paper. Here, we will response to all the comments one by one as follows:

**Major comments:**

(1) English expression needs further improvement, which is not bad but needs to be re-checked carefully. Some sentences are not clear enough.

Reply: We have followed the specific comments (see below) from the reviewer to make the expression clearer. Besides, we have also checked the whole manuscript several times to improve the words and sentences.

(2) A detail introduction of the vertical measurements of BC and its mixing state is missing. although the authors note that there are limited measurements, their results should be included for intact.

Reply: The vertical measurement about BC and its mixing state is limited and mainly conducted by aircraft. We have added more descriptions about BC's mixing state in the introduction and a brief summary about the vertical measurement about BC's mixing state in Table S2. (Line 51-53)

Limited vertical measurements of BC and its mixing state using the aircraft approach have been conducted (Katich et al., 2018;Schulz et al., 2019;Schwarz et al., 2013;Schwarz et al., 2017). Generally, BC concentration decreases with altitude, but the vertical distribution of size distribution and coating thickness could vary a lot due to the regional transportation (Hu et al., 2020), pollution levels (Ding et al., 2019;Zhao et al., 2020), biomass burning plumes (Ditas et al., 2018) and other factors. However, as summarized in Table S2, aircraft measurement mainly focuses on the vertical difference between boundary layer, upper troposphere and lower stratosphere. The data points in the boundary layer were limited due to flight height restrictions.

**Table S2 Detail timetable of the vertical profile measurement**

| Measurement area | Measurement description | BC conc. | BC core Size distribution | Coating thickness | Reference |
|---|---|---|---|---|---|
| Europe | Aircraft measurement (0-1000m), EUCAARI campaign. | ~6-200 ng/m3, decreasing with altitude. | With MMD of 150-210 nm, decreasing with altitude. | - | Ding et al., 2019) |
| Rural Beijing | Aircraft measurement (0-3000m). | ~50-3000 ng/m3, decreasing with altitude. | With MMD of 160-230 nm, the vertical profile of MMD varied from cases. | Higher coating thickness in the boundary layer under pollution conditions. | (Ding et al., 2019;Zhao et al., 2020) |
| Arctic | Aircraft | 1.4-50 ng/m3, generally | With MMD of 130-200 | - | (Schulz et al., |

| | | | | | |
|---|---|---|---|---|---|
| | measurement (0.1-5500m), NETCARE campaign | decreasing with altitude, but existing concentration peaks at certain height | nm, decreasing with altitude in Spring but uniform in summer. | | 2019) |
| Europe to North America | Aircraft measurement (2-20km) | 1-40 ng/m3, decreasing with altitude | - | Significant coating thickness increase during plume affected period | (Ditas et al., 2018) |
| Global | Aircraft measurement, HIPPO and Atom campaign | 1-10 ng rBC/kg air in the upper troposphere, 0.5-2 ng rBC/kg in the lower stratosphere | - | - | (Katich et al., 2018;Schwarz et al., 2013;Schwarz et al., 2017) |

(3) Some conclusions are not supportive enough. for instance, the authors mentioned that the boundary layer in summer is relatively high. Then, how much does 0-240 m take up in the boundary layer? In addition, the variation in the boundary layer leads to such vertical profile type changes in Fig.3, but without showing the variations of boundary layer data. It is necessary to include boundary layer height throughout the day to support the discussion.

(4) "In fact, the several gradual decrease cases observed during the daytime generally occurred on cloudy days when the radiation was relatively weak." I do not find such data in Fig.3. Supporting data should be provided.

Reply: We have added data of mixing layer height and cloud amount in the discussion to support our discussion.

1)  In the method section, we have added the sources of new data. (Line 100-104)

The height of boundary layer height (BLH) was determined (Tang et al., 2015;Tang et al., 2016) using the vertical profile of the attenuated backscatter coefficient measured by a single-lens ceilometer (CL51, Vaisala, Finland) at the same observation site. The cloud amount data was extracted from the reanalysis data product ECMWF Reanalysis v5 (ERA5, https://cds.climate. copernicus.eu/cdsapp#!/home) with spatial resolution of 0.25°*0.25°and time resolution of 1 hour.

2)  In Fig. 1, we have added the time series of BLH and cloud amount.

[Figure]

**Figure. 1** Time series of the major observed parameters during the observation; the data were processed to a 1-hour resolution. (a) The height of the moveable container. (b) Mass concentration of black carbon and absorption (babs) and scattering efficient (bsca), the light-coloured lines denote the data with 1-min resolution. (c) The coating thickness (Dp/Dc) and mass median diameter (MMD) of BC, the light-coloured lines denote the data with 1-min resolution. (d) The mixing ratio of O3 and NOx. (e)(f) The meteorological conditions during

the observation period.

3) We have mentioned how much does 0-240 m take up in the boundary layer in the result section. (Line 149-152)

The BLH ranged between 50-2000 m (Fig. 1f) during the observation. The variation of BLH followed a clear diurnal pattern (Fig. S1) with an average value of 400-500 m at night and 1000-1200 m at noon. The observation height (0-240 m) was generally in the mixing layer. However, occasionally at night and in the morning, the BLH could be below 240 m, and the moveable container could detect the BC properties in the residual layer in such cases.

[Figure]

**Figure. S1 Diurnal variation of BLH during the observation.**

4) We have referred Fig. S1 in the discussion of the diurnal change of vertical profile type. (Line 256-257)

The diurnal variation in the boundary layer (Fig. S1) leads to such vertical profile type changes.

5) The measured BLH was also used to discuss the sudden decrease type. (Line 212-220)

Lei et al. (2021) found that in the stratified type, the BLH measured by ceilometer was relatively low (~110-250 m), which is consistent with the findings of this study. The BLH measured by ceilometer in the stratified type ranged between 100-300 m, which is lower than the average value of ~400-500 m. On the night of 27th June, the BLH measured by ceilometer even reached 80 m. It is known that the absolute value of BLH determined from different methods can vary significantly. Wang et al. (2018a) found that in the stratified type, the BLH determined from ceilometer would always be higher than the sudden concentration transition point and suggested that it is more precise to determine the BLH by using the sudden concentration transition point in such cases. In this study, the top of the BLH was regarded as the sudden concentration transition point in the sharp decrease type, and the profiles were used to investigate the differences in pollutants between the night boundary layer and residual layer in the following section.

6) The cloud data was used to discuss the gradual decrease type during the daytime. (Line 261-264)

In fact, several gradual decrease cases observed during the daytime, such as 11:00-12:00 on 2nd July (Fig. S13) and 19:00 on 27th June (Fig. S8), generally occurred on cloudy days when the radiation was relatively weak. As shown in Fig.1, the total amounts of clouds were nearly 100% for the two cases and the radiation was relatively low compared with that of the entire observation except several rainy days.

**Specific comments:**

1. Line 37: The expression of "the area" is not good. "the surrounding atmosphere" may be better.

Reply: Thanks for the reviewer. We have followed the advice. (Line 38)

Such depressing of the boundary layer is called the "dome effect" in which the BC in the upper boundary layer could heat the surrounding atmosphere, increasing the stability of the boundary layer.

2. Line 42: Change "study" to "studies".

Reply: Thanks for the reviewer. We have followed the advice. (Line 44)

The present studies mainly focused on the vertical distribution of BC mass concentrations (Wang et al., 2018a;Samad et al., 2020;Wu et al., 2021a;Guan et al., 2022).

3. Line 50: "Shiraiwa et al. (2010) found that BC's absorption ability after coating could reach two times greater than BC without coating." The relationship between BC's absorption ability and coating depends on many factors, the description here is not appropriate since the atmospheric conditions or particle properties are missing.

Reply: We have changed the sentence to make it clearer. (Line 51-53)

Through laboratory study, Shiraiwa et al. (2010) found that BC's absorption ability after coating could reach two times greater than BC without coating. Such absorption enhancement of BC due to coatings was also observed in the ambient with typical factors of 1.0-1.5 (Liu et al., 2017).

4. Line 59: What properties of BC has been measured? Concentration? Please describe clearly.

Reply: This sentence has been extended (See the reply for major comments). (Line 62-68)

5. Line 88: Please describe clearly again. The sentence should be "The concentration of pollutant gases (…), the concentration and mixing state of BC, and optical properties (…) of aerosol were measured…".

Reply: Thanks. The sentence has been changed. (Line 95-96)

The concentration of pollutant gases (NO2, O3, CO), the concentration and mixing state of BC and optical properties (light extinction and absorption) of aerosol were measured

6. Section 2.2: Eabs is calculated at a wavelength of 550 nm, but the Dp used in the calculation of Eabs is calculated at a wavelength of 1064 nm. Does this lead to higher uncertainty in the calculation of Eabs? Please describe it with more details.

Reply: The wavelength SP2's laser is 1064 nm. Thus, the $D_p$ is derived at a wavelength of 1064 nm. We think the uncertainty in Eabs calculation at 550 nm is mainly caused by two factors:

1) The refractive indices

The refractive indices of BC are wavelength dependent. In the determination of $D_p$, we used the 2.26-1.26i at wavelength of 1064 nm recommended by (Moteki and Kondo, 2008). In the

calculation of Eabs at 550 nm, we used the 1.95+0.79i recommended by (Bond and Bergstrom, 2006). We have mentioned the chose of refractive indices in the method section. (Line 111 and Line 140-141)

2) The morphology

Since the wavelength of 1064 nm is much larger than BC's diameter, the optical properties of BC is less sensitive to particles' morphology. Liu et al. (2015) found the bias of core-shell assumption in $D_p/D_c$ determination would be less than 6%. However, for the calculation of Eabs at 550 nm, the morphology may play a more important role in BC's optical properties.

We have added the discussion of uncertainties from morphology. (Line 112-113 and Line 141-144)

7. It is mentioned that the core-shell assumption will lead to overestimation of light absorption. Is the light absorption shown in the result section corrected?

Reply: Since we didn't obtain the morphology information data of BC during this observation and the uncertainties of BC optical properties due to morphology may differ among different cases, we didn't correct the $E_{abs}$ in result section. However, we have informed the readers about the extent of this overestimation by citing references. We would clarify it in the method section. (Line 141-144)

The Eabs may be overestimated in this study due to the core-shell assumption in the calculation (Liu et al., 2017;Wu et al., 2018;Liu et al., 2019b) because of the complex morphology of BC (Wang et al., 2017;Hu et al., 2021).

8. Line 139: Reword to "…, which is consistent with…".

Reply: Thanks for the advice, we have changed the expression. (Line 159)

The mass size distribution of the BC core followed a lognormal distribution with a mass median diameter (MMD) of 173 nm and a geometric standard deviation (GSD) of 1.57, as shown in Fig. S2, which is consistent with the MMD value (171 nm) observed in Beijing in the summer of 2018 (Liu et al., 2020b).

9. Lines 148-150: As mentioned above, June 25 is an ozone pollution day with a maximum hourly mixing ratio of 124 ppb. From figure 1, the ozone mixing ratio is about 40 ppb on July 5. Is this the ozone pollution day?

Reply: Thanks for pointing our mistake, it's a typo. The ozone pollution day is June 25, the $D_p/D_c$ increased from 1.1 to 1.4 at that day. We have corrected it. (Line 169)

It is also noted that the Dp/Dc value could increase quickly from 1.1 to 1.4 during the ozone pollution day (25th June).

10. Lines 174-175: From figure S12, the concentration of O3 increase with increasing height in the vertical profile at 18:00 on July 2, which is not consistent with the description here.

Reply: It's another typo and also pointed by another reviewer. It should be 'increase' but not

'decrease', we have changed it. (Line 198-199)

The concentration of O3 increased with increasing height at a sharper slope than that in the uniform case.

Reply: Thanks, we have added the references. (Line 202)

This type is also referred to as the 'stratified type' in other studies (Guimaraes et al., 2019;Lei et al., 2021).

12. Lines 203-206: Please check carefully. The figure does not match with the text here. Figure S9 is not the vertical distribution on July 9. Moreover, the vertical distribution at 11:00 on July 9 is not sudden increase type in concentration of BC like the vertical profile at 11:00 on July 11. And, how do you figure out that the sudden increase in this case is due to solid fuel burning? Is there any other evidence here?

Reply: The vertical profile at 11:00 on July 9th is regarded as the sudden increase type in this study. It should be Fig. S16, we have corrected the text.

At 11:00, the BC concentration near the surface (0-50m) was significantly higher than at mid-air, and was characterized by an obviously thin coating and large MMD (denoted by the red circle). This suggests that the vertical profile at 11:00 was influenced by a fresh plume. Because there is no significant $NO_x$ concentration increase and MMD increased instead of decreasing in this case. We thought the plume may be originated from solid fuel burning.

We have changed the expression to make it clearer. (Line 233-236)

The sudden increase that occurred at 11:00 on the 9th of July, as shown in Fig. S16, was accompanied by an increase in MMD, a decrease in Dp/Dc and no significant variation in NOx concentration. The sudden increase in this case may be influenced by the emission from solid fuel burning, since the MMD is generally higher for BC from solid fuel burning than that of fossil fuel burning (Schwarz et al., 2008;Pan et al., 2017;Liu et al., 2019a).

[Figure]

**Figure S16 Vertical profiles during 9th July**

13. Lines 208-209: Reword to "…the vertical profiles of BC, including its concentration and mixing state,…".

Reply: Thanks, we have modified the expression. (Line 239)

Fig. S7 shows the profiles on the ozone pollution day (maximum hourly ozone mixing ratio reached 124 ppb), the vertical profiles of BC, including its concentration, mixing state and ozone presented in a uniform type.

14. Lines 250-251: The figure does not match with the text again. From figure 4, the vertical profile of Dp/Dc at 23:00 slightly increase with height above 160 m. So the following explanation might be incorrect?

Reply: It's a typo, we have corrected it. Many thanks for the reviewer. (Line 283)

The Dp/Dc is nearly uniform below 160 m and slightly increases with height above 160 m.

15. Line 262: Reword to "…than that in the residual layer."

Reply: It has been reworded. (Line 295)

The BC properties, O3 and NOx at the 240 m height and ground are counted for the 5 sharp decrease profiles to study the pollutant difference between the ground and residual layers. For BC aerosols, the concentration is much higher on the ground than that in the residual layer.

16. Line 263: It is better to change "more coatings" to "thicker coating".

Reply: Thanks, we have followed the advice. (Line 296)

The BC concentration on the ground was 84.6% higher than that in the residual layer at 23:00 on the 8th of July. BC in the residual layer generally had thicker coatings, indicating a higher aging degree.

17. Lines 268-269: Note the preciseness of the data here. The MMD described here is inconsistent with that shown in the figure S7.

Reply: Thanks for the advice. We have changed it. (Line 301-302)

The MMD values were relatively stable (~170-190 nm) on the ground, while the MMD in the residual layer was more variable, ranging between 210 nm and 165 nm.

18. Lines 275-276 and 279: It should be "at 23:00 on June 27,…".

Reply: We have corrected it. (Line 309-312)

19. Line 304: Reword to "…at the top of the boundary layer during the daytime…".

Reply: We have changed the expression. (Line 336)

the pollutants from the residual layer firstly influence at the top of boundary layer during the daytime denoted by the large difference of O3 and Dp/Dc between 240 m and the surface at 6:00.

20. Line 343: Dp/Dc represents the relative thickness of the coating of BC-containing particle. But, Dp/Dc is the diameter ratio of BC-containing particle to BC core, which is not shell to core.

21. The conclusion section is really tedious, mostly repeating the results. I suggest the authors focus on new findings.

Reply: Thanks for the reviewer. We have rewritten the conclusion section. (Line 378-411)

**References**

Bond, T. C., and Bergstrom, R. W.: Light absorption by carbonaceous particles: An investigative review, Aerosol Science and Technology, 40, 27-67, 10.1080/02786820500421521, 2006.

Liu, D. T., Taylor, J. W., Young, D. E., Flynn, M. J., Coe, H., and Allan, J. D.: The effect of complex black carbon microphysics on the determination of the optical properties of brown carbon, Geophysical Research Letters, 42, 613-619, 10.1002/2014gl062443, 2015.

Moteki, N., and Kondo, Y.: Method to measure time-dependent scattering cross sections of particles evaporating in a laser beam, J Aerosol Sci, 39, 348-364, 2008.

---

## Author Comment (AC2)

**Reply to the comments of anonymous reviewer #3 on manuscript Entitled " Vertical distribution of black carbon and its mixing state in urban boundary layer in summer"**

We sincerely appreciate the patient and insight comments and recommendations of the reviewer in improving this paper. Here, we will response to all the comments one by one as follows:

**General Comments:**

The authors can give more discussion on the significance of the work. In earlier studies, what had been left open, and what the present paper now to address? How their finding are relevant to atmospheric chemistry and radiative transfer? What the novel insights of their work are?

Reply: Thanks for the advice from the reviewer, we have rewritten the conclusion part to make the significance of the work clearer. (Line 405-411)

**Specific comments:**

(1) Abstract. Please indicate where the measurements are taken in the abstract and the date range of the measurements.

Reply: We have added the detailed information about the measurement location and time in the abstract. (Line 17)

In this study, vertical measurements were conducted through a moveable container based on a meteorology tower in Beijing urban area during June and July.

(2) SP2 data analysis: The minimum core size (from LII) and minimum total particle size (from scattering) differ, with the latter greater than the former. Have the authors filtered to account for any mismatch in the SP2 size for incandescence vs scattering, which can bias results? What are the limits on the Dp? When calculating BC concentration calculation, have the authors considered the SP2 limitations?

Reply: It is worth noting that the size range of pure scattering particles (~170-500 nm) differs from that of BC particles (~70-600 nm) due to the distinct responses of scattering and incandescence (LII) signals to particle size. This detection size range has been reported in the instructions of the SP2 instrument and in previous studies (Schwarz et al., 2008).

Regarding the BC core size (Dc), only the LII signal was used, so there is no mismatch between the LII and scattering signals.

The scattering signal was used to derive the optical diameter of BC-containing particles (Dp) through the LEO-fitting method. However, due to uncertainties in refractive indices, BC morphology, and the fitting area in the LEO method, Dp can sometimes be lower than Dc. We believe this is the mismatch referred to by the reviewer. In such cases, we retained the Dp derived from the LEO-fitting method. Although the Dp/Dc ratio may be lower than 1 for some individual particles, which is not physically correct, it did not affect the bulk characteristic of Dp/Dc. This mismatch has also been reported in previous studies(Laborde et al., 2013;Liu et al., 2019).

The calibration material and limited detection size range can affect the determination of BC concentration. To address this issue, we used the log-normal function to fit the BC size distribution and determine the

missing fraction of BC mass concentration. This missing fraction was then used to calibrate the results. In the Method section (lines 110-117), we have described the major parameters such as refractive indices, calibration material, calibration compensation factor, and so on. Introducing additional parameters related to Dp and BC concentration determination would make the section too long, so we refer readers to our previous study (Liu et al., 2020) for more detailed information.

(3) Page 5/Line 149-151: The authors are trying to add some discussions on the causes of BC coating growth during ozone pollution day. However, there is no sufficient evidence to support. The authors can support their explanation by previous studies, e.g.https://doi.org/10.1021/acs.est.2c00090.

Reply: The reference provided by the reviewer mainly reported the influence of $O_3$ and atmospheric oxidation ability on BC's aging, which is well relevant to the discussion in this part. Thanks for the reviewer, we have added this reference to support our argument and extended our discussion. (Line 169-173)

Previous studies have suggested that BC aging could occur more rapidly in the presence of increased O3 concentrations (Zhang et al., 2022). Furthermore, Liu et al. (2020b) found the Dp/Dc could increase at a faster rate during the daytime under high Ox condition. These findings suggest that severe photochemical processes and high atmospheric oxidation rates could contribute to the aging of BC and the subsequent increase in its coating thickness.

(4) Page 6/Line 165-166: The author can give NO data rather than NOx to demonstrate titration reaction.

Reply: Thanks for the reviewer, we have now changed the expression $NO_x$ to NO. (Line 185-187)

The presence of substantial NO emissions from traffic sectors would decrease the concentration of ozone through the titration reaction between ozone and NO, leading to the decreasing trend of ozone concentration with height.

(5) Page 6/Line 173-174: The author can discuss the emission sources of the measured BC based on MMD.

Reply: We have added more discussion about BC's core size to make the discussion clearer. (Line 194-197)

Studies have shown that the size of the BC core varies depending on the emission source. For instance, BC emitted from traffic sources typically has a lower MMD of about 150-180 nm, while that from solid fuel burning sources has a higher MMD of about 170-230 nm (Schwarz et al., 2008;Pan et al., 2017;Holder et al., 2014). In Beijing and London, (Liu et al., 2014;Liu et al., 2019a) found that the MMD of BC was much larger in winter than in summer, likely due to increased solid fuel burning for heating purposes.

(6) Page 6/Line 175-176: O3 decreased with increase height? I think that should be "O3 increased with ⋯⋯".

Reply: Thanks for the reviewer, it's a typo. It should be "$O_3$ increased with ⋯", we have modified the mistake. (Line 200)

The concentration of O3 increased with increasing height at a sharper slope than that in the uniform case.

(7) Page 7/Line 296-308: The measurement found higher O3 and Dp/Dc in the residual layer. The authors can give more discussion on BC aging in the residual layer. How the finding is relevant to atmospheric chemistry and radiative transfer?

Reply: We have added more discussion in this part. (Line 341-344 and Line 349-350)

The existence of aerosol, especially the absorbing aerosols, would depress the development of mixing layer (Ding et al., 2016;Wang et al., 2018b). The higher Dp/Dc of BC in the residual layer may amplify the absorbing of BC in the upper height in the early mornings, resulting in a more stable structure and deteriorating the air pollution.

Besides the high actinic flux, the higher O3 concentration may also contribute to the Dp/Dc increase in the upper height.

(8) Conclusions: The authors need to clear the novel insights of their work and the significance of their findings.

Reply: The conclusion part has been rewritten. (Line 379-412)

**Reference**

Laborde, M., Crippa, M., Tritscher, T., Juranyi, Z., Decarlo, P. F., Temime-Roussel, B., Marchand, N., Eckhardt, S., Stohl, A., Baltensperger, U., Prevot, A. S. H., Weingartner, E., and Gysel, M.: Black carbon physical properties and mixing state in the European megacity Paris, Atmospheric Chemistry and Physics, 13, 5831-5856, 10.5194/acp-13-5831-2013, 2013.

Liu, D. T., Joshi, R., Wang, J. F., Yu, C. J., Allan, J. D., Coe, H., Flynn, M. J., Xie, C. H., Lee, J., Squires, F., Kotthaus, S., Grimmond, S., Ge, X. L., Sun, Y. L., and Fu, P. Q.: Contrasting physical properties of black carbon in urban Beijing between winter and summer, Atmospheric Chemistry and Physics, 19, 6749-6769, 10.5194/acp-19-6749-2019, 2019.

Liu, H., Pan, X. L., Liu, D. T., Liu, X. Y., Chen, X. S., Tian, Y., Sun, Y. L., Fu, P. Q., and Wang, Z. F.: Mixing characteristics of refractory black carbon aerosols at an urban site in Beijing, Atmospheric Chemistry and Physics, 20, 5771-5785, 10.5194/acp-20-5771-2020, 2020.

Schwarz, J. P., Gao, R. S., Spackman, J. R., Watts, L. A., Thomson, D. S., Fahey, D. W., Ryerson, T. B., Peischl, J., Holloway, J. S., Trainer, M., Frost, G. J., Baynard, T., Lack, D. A., de Gouw, J. A., Warneke, C., and Del Negro, L. A.: Measurement of the mixing state, mass, and optical size of individual black carbon particles in urban and biomass burning emissions, Geophysical Research Letters, 35, Artn L13810 10.1029/2008gl033968, 2008.

---

## Author Response (AR2)

Dear editor,

Thank you for checking our manuscript, and we apologize for the errors in the revised version. We have carefully reviewed and corrected any grammar and terminology issues.

Additionally, we have included Professor Guiqian Tang (tgq@dq.cern.ac.cn) as a co-author for his contributions to the boundary layer height data and relevant discussion in the manuscript.

Sincerely yours,

Hang Liu

Please double-check for the right format for citations in the text. Currently, there are no blancs between different references, for instance line 35: "(Ramanathan and Carmichael, 2008;Bond et al., 2013;Li et al., 2022)".

Reply: We have added spaces between different references.

Please double-check for proper use of articles and other grammar issues, for instance line64: "Generally, the BC concentration decreases …"

Reply: We have carefully reviewed the grammar issues again. The changes are highlighted below.

Caption of Figure 1: First sentence: repeats "observed" and "observation"; do you mean 1-hour "average" instead of "resolution", and was it a running average or other? Please specify and explain either here or in the text. Second sentence: "coefficient", not "efficient". Please double-check the manuscript once more in particular for the proper use of technical terms.

Reply: Thanks to the editor, we have made changes to the sentences in the caption of Figure 1. The resolution and average method are listed in Table S3. We have corrected the misuse of "efficient" throughout the manuscript, including the caption of Figure 2.

Line 149: What do you mean with "vertical information"? I think it is more correct to say "Thus, the time series in Fig. 1 also include information on the vertical distribution of pollutant concentrations.
Line 342: "The presence of aerosol …" instead of "existence".

Reply: Thanks to the editor. We agree that the suggested sentence/word is more correct. We have made the necessary changes.

Table S1: I recommend removing the word "Detail" from the caption, or to say "Detailed timetable …"

Table S2: Table caption is the same as for Table S1, but the table has different content. Please modify the caption accordingly.

Reply: We have followed the advice. The captions of Table S1 and Table S2 are listed below:

**Table S1 Timetable of the vertical profile measurement**

**Table S2 Brief summarize of vertical measurement about BC and its mixing state**

[revised manuscript text omitted]